

# Paw-spective shift: how our mood alters the way we read dog emotions

Holly G. Molinaro and Clive D.L. Wynne

Psychology Department, Arizona State University, Tempe, AZ, United States of America

## ABSTRACT

This study explored the influence of people's mood on their perception of dog emotions in order to expand our understanding of how mood biases may shape emotional interpretation. Across two experiments, participants were primed into positive, neutral, or negative moods using validated visual stimuli before they evaluated video clips of dogs displaying positive, neutral, or negative emotional states. Participants were asked to rate valence and arousal of the dogs in the videos. Experiment 1 utilized visual primes unrelated to animals, while Experiment 2 employed dog-specific primes. Although mood priming significantly influenced participants' self-reported emotions in Experiment 1, it did not affect their interpretation of the dogs' emotional states. Dog-specific primes influenced participants' interpretation of dog emotions in Experiment 2; however, the effect was a contrasting one, in that participants in the positively induced group rated dogs as sadder and those in the negative group rated dogs as happier. These findings challenge previous assumptions about mood-congruence effects in cross-species emotional perception, suggesting a more complex interplay of factors than anticipated. The study underscores the need for further research to disentangle the mechanisms governing how humans perceive and respond to animals' emotional cues, with implications for improving animal welfare and human-animal interactions.

## INTRODUCTION

Our understanding of dogs' emotional states may be more multiply-determined—and often less accurate—than we realize. How we perceive our dog's, or any other animal's, emotional state can have lasting consequences for how we care for them. While the study of animal emotions is valuable in itself, how humans perceive and interpret these emotional cues has practical implications. Folk wisdom indicates that a dog wagging its tail is happy, and a dog with its tail between its legs is sad. Scientific findings are more varied, with some research indicating we can easily read dog emotions from their facial expressions or behavior (*Bloom & Friedman, 2013*), while other studies point to limitations and biases in our interpretations (*Molinaro & Wynne, 2025*). The aim of the research program of which this paper forms a part is to uncover the biases that shape everyday observations of dog emotions and develop more accurate methods for understanding and improving animal care across various contexts.

Although research on the perception of dog emotions is limited, valuable insights can be gleaned from understanding how humans perceive emotions in one another. This literature

Corresponding author
Holly G. Molinaro,
hgmolina@asu.edu

reveals that a variety of factors influence people's ability to interpret others' feelings. Whereas facial expressions and body language provide crucial information (*Meeren, Van Heijnsbergen & De Gelder, 2005*; *Manstead, Wagner & MacDonald, 1983*), contextual elements such as the situation at hand, personal experiences, cultural background, and even one's own current mood (*Barrett et al., 2019*; *Hess & Hareli, 2016*) also influence emotion perception. These multiple factors collectively shape our emotional perceptions, and highlight the complexity of interpreting emotions, whether in humans or animals.

The distinction between 'emotion' and 'mood' is still debated in the literature (see *Price, 2024*). In general, mood is a longer-lasting affective state not caused by any one specific stimulus, while emotion is a briefer affective state induced by certain stimuli or events and more intense than a mood (*Beedie, Terry & Lane, 2005*). In other words, mood can be described as how one is presently feeling. In humans, it has been found that how one is feeling in the moment affects many different cognitive processes, including the perception of other's emotions. Mood impacts how we perceive our own cognitive processes (*Marino et al., 2009*), negative mood impacts academic performance of children (*Scrimin, Mason & Moscardino, 2014*) and mood even affects our 'theory of mind' skill, in that those in a happy mood may be less likely to use this ability (*Converse et al., 2008*).

Mood has been found to have diverse impacts on the perception of emotions (*Gross & Feldman Barrett, 2011*; *Lench, Flores & Bench, 2011*; *Zadra & Clore, 2011*). One study had three different groups of participants watch sad, happy or neutral video clips before they had to recognize either happy or sad faces. It was found that emotion perception was more difficult when participants viewed a facial expression not congruent with how they currently felt (*Schmid & Schmid Mast, 2010*). Another study found similar results: participants who were primed into a happy mood perceived happy expressions to be present on a neutral face for a longer time than sad expressions (*Niedenthal et al., 2000*). Both studies evidence the emotional congruence effect, whereas one is more likely to perceive the emotion one is experiencing oneself. This effect was also found with older participants, where those in a sad mood rated facial expressions of happiness as sadder compared to participants not in a sad mood (*Lawrie, Jackson & Phillips, 2019*).

While it is well-established that mood influences our perception of human emotions, research on how this applies to the perception of emotions in animals is still scarce. The interpretation of emotions in animals—distinct from the actual emotions they may be experiencing—can be shaped by a variety of factors. Women have been found to be more likely to perceive emotions such as grief or love in animals compared to men (*Walker et al., 2014*). In addition, how familiar we are with a species also impacts our perception of emotional states, in that humans are better at recognizing emotions of dogs compared to tree shrews (*Scheumann et al., 2014*). When it comes to dogs specifically, most research indicates that humans can readily perceive their emotions; however, several factors can influence this ability. For example, children have been found to be worse at perceiving dog emotions than adults (*Lakestani, Donaldson & Waran, 2014*). Furthermore, people's cultural background can influence their ability to perceive dog emotions. Individuals raised in cultures where dogs are not commonly kept as pets may struggle with interpreting canine emotions compared to those who grew up in households with dogs (*Amici et al., 2019*).

Finally, dog experience appears to have mixed effects, in that some studies show experience with dogs positively impacts perception of emotion (*Bloom & Friedman, 2013*; *Wan, 2011*) while other studies indicate the opposite effect (*Pongrácz, Molnár & Miklósi, 2006*; *Tami & Gallagher, 2009*). Owner attachment could also influence perception of emotion in one's pets, as previous research has shown that the degree of emotional attachment individuals feel toward their companion animals is associated with greater attribution of complex emotions to those animals (*Arahori et al., 2017*; *Morris, Doe & Godsell, 2008*).

A recent study indicated that perception of dog emotions may be less accurate than previously believed. It was found that people's perception of dog emotions depended more on extraneous aspects of the situation the dog was in than on the behavior of the dog itself (*Molinaro & Wynne, 2025*). Therefore, the aim of this current study was to elucidate what other factors could influence human perception of dog emotions. Specifically, we investigate here whether the mood a participant is in impacts how they perceive dogs are feeling, similarly to what has been observed in the human literature of the effects of mood on emotional perception. Misreading or overlooking emotional cues can lead to inappropriate handling, delayed intervention, or unmet behavioral and psychological needs for animals in human care. By better understanding how human emotional state may bias these perceptions, we can improve human-animal interactions and support more accurate, empathetic, and welfare-conscious care.

We hypothesized, based on the emotional congruence effect, that humans in a positive mood would be more likely to view dogs as in a positive mood, and conversely for a negative mood. Ultimately, we hoped to better understand the ways in which we perceive our canine companions, in order to better care for them in our home and shelter environments. In addition, this research can also have lasting impacts for future studies of perception of other animal's emotional states.

# EXPERIMENT 1
# MATERIALS & METHODS

The aim of Experiment 1 was to test the impact of mood induced with standard visual materials on the perception of emotion in videos of dogs that had been induced into positive, neutral, and negative emotional states.

### Filming of dog videos

Three dogs were used. Each dog was filmed with its owner on a neutral background, with stimuli used to elicit positive, negative, and neutral emotional states. All videos were recorded an iPhone 13 Pro Max (Apple Inc., Cupertino, CA).

The first dog, Oliver, was a 14-year-old medium-sized mixed-breed dog (approximately 30 kg). Video recordings of Oliver were already used in *Molinaro & Wynne (2025)* and videos of Oliver in positive, negative, or neutral situations were used again in this study. The second dog, Canyon, was a 1-year-old medium size Catahoula dog (approximately

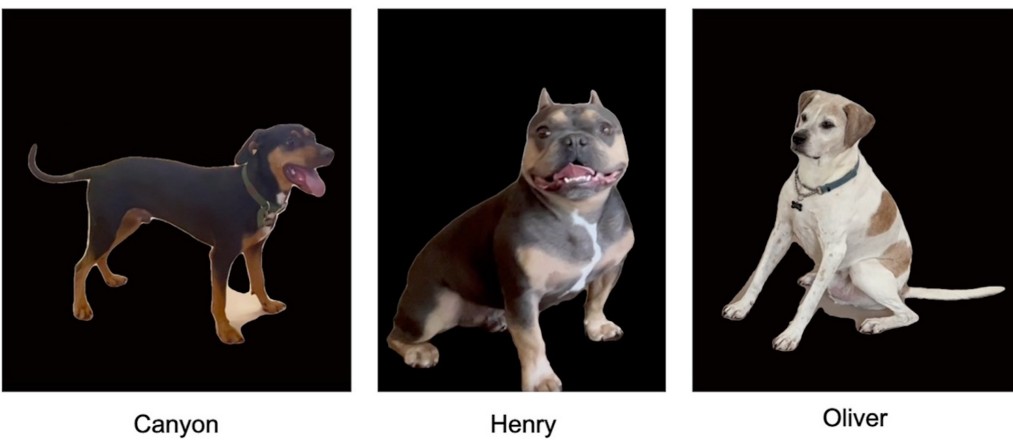

Canyon          Henry          Oliver

**Figure 1** The three dogs used in the dog videos.

30 kg). The third dog, Henry, was a 3-year-old small size French bulldog (approximately 10 kg). Photos of the three dogs are in Fig. 1.

Each dog was presented with stimuli to elicit positive or negative emotional states for that dog based on discussion with the dogs' owners. Videos were also taken of the dog when he was resting or not actively engaging with a stimulus and these were considered as a neutral state. Altogether, three videos were used of each dog; one in an assumed positive state, one in an assumed negative state, and one in an assumed neutral state. All videos are available in the Supplementary Materials.

## Video editing

iMovie software (Apple Inc., Cupertino, CA, USA) was used to first edit all videos clips so that only the interaction between the dog and the owner and other stimuli was shown. These video clips lasted between eight and 30 s. Next, the clips were edited in Adobe After Effects (Adobe Systems Inc., San Jose, CA, USA) software to remove the background, so that only the dog was visible on a black background. Any sound from the dog was retained, but any other sound was removed. Some video clips were then shortened again so that all nine videos were between eight and 15 s long. The positive stimuli were being shown a treat (Oliver), being shown a toy (Canyon) and being told he was seeing "Grandma" (Henry). The final negative stimuli were being shown a cat (Oliver) and being shown a vacuum cleaner (Canyon and Henry). The final neutral videos were all videos where the dog was resting or waiting for the owner to grab another stimulus. There were a total of nine videos, with each of the three dogs being shown in a positive, neutral, or negative state.

## Priming photos

Images from the Nencki Affective Picture System (NAPS) were used to induce positive, neutral, or negative emotional states for human participants taking the survey. Images have been found to be most effective at inducing emotional states (*Lench, Flores & Bench, 2011*). NAPS images have been validated to induce both discrete emotions and as well as

increase or decrease valence and arousal (*Marchewka et al., 2014*; *Riegel et al., 2016*). The NAPS image database is split into five categories: animals, faces, landscapes, objects, and people. The category of animals was not used in this study as we wanted to investigate human and animal priming materials separately.

Valence rankings from *Marchewka et al. (2014)* were used to select images for positive, neutral, and negative emotional priming. Thirty to fifty images positioned in the bottom third, middle third, and top third of valence rankings were initially selected by the first author for the negative, neutral, and positive inducing images. Seven images from each class were then selected so there were at least one or two images from each of the four categories (faces, landscapes, objects, and people).

### Survey

Qualtrics (Qualtrics, Provo, UT, USA) software was used to create the survey. There were three separate versions of the same survey, with one having the positive inducing images, one having the neutral inducing images, and one having the negative inducing images. Participants click an arrow on screen to begin the survey and indicate their consent and agreement. The first block of the survey consisted of demographic questions (age, gender, ethnicity) as well as dog experience level. The second block showed participants the seven images, where they were asked to clear their mind and think about how the images make them feel. This was followed by a prime manipulation check where participants rated on a 7-point Likert scale how they were feeling after viewing the images (0–7). Manipulation checks are essential for verifying the causal impact of the intended variables, and using explicit mood measures is one of the most effective ways to confirm that the desired effect has been achieved (*Ejelöv & Luke, 2020*). In the next block of the survey participants viewed the nine videos. The video order cycled through the positive, neutral, and negative emotional state of the dogs, so that the first three videos showed dog Canyon, the second three showed Henry, and the final three showed Oliver. We grouped all three clips of each dog together to help participants form a more coherent impression of each individual dog. For each video, participants answered a free response question on what they thought the dog was feeling, while also rating the dog's valence (how good or bad they thought the dog was feeling) and arousal (how calm or agitated they thought the dog was feeling) on a scale of one to 10. Halfway through watching the videos, a check in question was asked, where participants had to pick a specific number from a list. This was done to ensure participants were paying attention to the stimuli. The final survey block was a single question asking for their ending mood, on a 7-point Likert scale.

### Participants

Arizona State University (ASU) Psychology undergraduates served as participants for this study. ASU Institutional Animal Care and Use Committee granted this study an exemption from review. ASU's Institutional Review Board for human subjects approved this research project on 8/18/2023 (STUDY00018423). The survey was launched on 9/12/2023 and ended when 300 respondents had completed the survey, 100 per each emotional condition version of the survey. We conducted an a priori power analysis using G*Power to determine

the necessary sample size. Based on a medium effect size ($f = 0.25$), an alpha level of 0.05, and power set at 0.95 for detecting group differences across three conditions using one-way ANOVA, the recommended sample size was approximately 252 participants. To ensure sufficient power and allow for potential exclusions (*e.g.*, failed check in questions or incomplete responses), we aimed for 300 total participants—100 per priming condition.

## Data analysis

IBM Statistical Package for the Social Sciences, SPSS (Version 28.0.1.0, IBM Corp., Armonk, NY, USA) was used to analyze all data. Data were visually assessed and were approximately normally distributed and displayed homogeneity of variance. Results were considered significant at an alpha level $p < 0.05$. One-way ANOVAs were used to compare the prime mood check differences between the three different primed groups, as well as the differences between the ending mood reported among the three different primed groups. One-way ANOVAs were also used to compare average valence and average arousal ratings among the three different primed groups. A paired t test was used to compare the prime mood check and ending mood reported for each of the three different primed groups. A between subjects factorial multiway ANOVA was used to examine the effects of demographic variables on the average valence, average arousal and prime check mood.

*Post hoc* comparisons using the Tukey HSD test were carried out on significant ANOVA results. A Kruskal Wallis test nonparametric ANOVA was used to compare free responses coded for valence, arousal, anthropomorphism, mentalizing, and action across the three different primed groups.

## Free response coding

Free response coding followed the same protocol from *Molinaro & Wynne (2025)*. Each free response was coded for five dimensions:

1. Valence (ranging from −2 to +2)
2. Arousal (ranging from −2 to +2)
    These first two were chosen to replicate the valence and arousal scale questions.
3. Presence or absence of anthropomorphism (1, 0)
4. Presence or absence of mental state terms (1, 0)
5. Presence or absence of action state terms (1, 0).

Two research assistants, unaware of the study's goals, initially coded a subset (150) of free responses. They reached a consensus on how to score each response for the five dimensions ("consensus coding", *Richards & Hemphill, 2018*).

Finally, the coders individually assessed another 20% of the free responses. Cronbach's alpha assessed interrater reliability and showed high consistency across all subcodes (valence, $\alpha = .981$; arousal, $\alpha = .982$; anthropomorphism, $\alpha = .972$; mentalizing, $\alpha = .964$; action, $\alpha = .947$). The two coders then individually scored all 2,619 free responses for the five dimensions listed above (see Supplementary Materials for codebook).

| Table 1 | Experiment 1 and 2 demographics. | | | | | |
|---|---|---|---|---|---|---|
| | **Negative prime** | | **Neutral prime** | | **Positive prime** | |
| | Exp #1 | Exp #2 | Exp #1 | Exp #2 | Exp #1 | Exp #2 |
| **Age Class** | | | | | | |
| 18–21 | 96% | 95% | 94% | 94% | 95% | 98% |
| 21+ | 4% | 5% | 6% | 6% | 5% | 2% |
| **Gender** | | | | | | |
| Female | 60% | 55% | 67% | 60% | 59% | 49% |
| Male | 34% | 44% | 33% | 40% | 40% | 49% |
| Other | 5% | 1% | | | 1% | 1% |
| **Ethnicity** | | | | | | |
| White | 45% | 53% | 52% | 65% | 53% | 51% |
| Hispanic | 19% | 21% | 18% | 15% | 22% | 25% |
| Asian | 28% | 14% | 19% | 10% | 14% | 14% |
| Other | 8% | 12% | 11% | 10% | 11% | 10% |
| **Dog experience** | | | | | | |
| Somewhat | 24% | 14% | 23% | 14% | 16% | 13% |
| Familiar | 14% | 22% | 12% | 14% | 16% | 19% |
| Familiar currently | 10% | 9% | 7% | 9% | 13% | 10% |
| Very | 52% | 55% | 58% | 63% | 55% | 58% |

# RESULTS

## Response inclusion criteria and demographics

A total of 100 participants took each of the surveys. Once those who did not answer the check in question were removed from the dataset, there were 96 participants for the negative priming group, 97 for the neutral and 98 for the positive.

Demographic descriptions are in Table 1 for the three prime group condition surveys. Because of the highly skewed age distribution of participants, age class was combined for analysis into two groups, 18–21 and over 22 years. Gender was combined into three groups (male, female, other). Ethnicity was combined into four groups (White, Hispanic, Asian, Other). Finally, dog experience was combined into four groups (None/Somewhat, Familiar, Familiar Currently and Very/Extremely Familiar).

## Priming and ending mood

Beginning prime mood check was significantly affected by the priming photos ($F_{2,288} = 456.73$, $p < .001$, $\eta p^2 = .76$, Fig. 2), showing the manipulation check was confirmed. In the negatively primed group, mood significantly increased from the starting mood prime check to the end of the survey ($t_{93} = 23.44$, $p < .001$, $d = 2.42$, Fig. 3). In the neutral primed group, mood significantly increased from the starting mood prime check to the end of the survey ($t_{96} = 7.67$, $p < .001$, $d = 0.78$, Fig. 3). In the positive prime group, mood significantly decreased from the starting mood prime check to the end of the survey ($t_{97} = -4.31$, $p < .001$, $d = 0.44$, Fig. 3). Ending mood was not affected by the prime group condition ($F_{2,288} = 2.29$, $p > .05$).

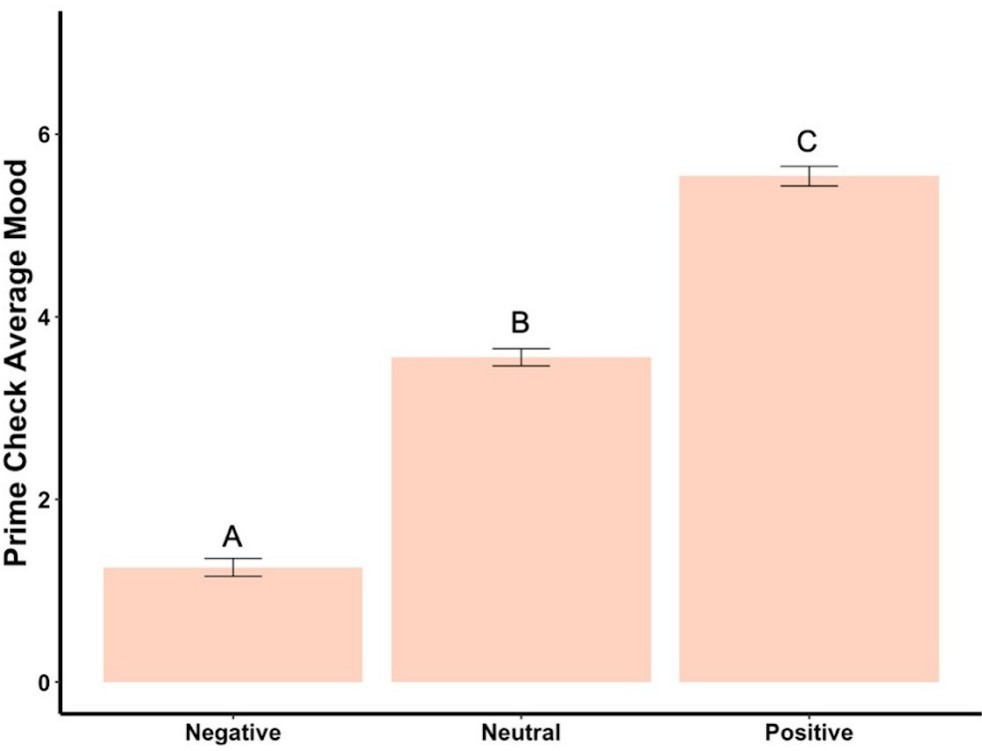

**Figure 2  Prime check mood averages across the three primed groups—Experiment 1.** Note: Different letters denote significant differences between conditions. Error bars show standard errors.

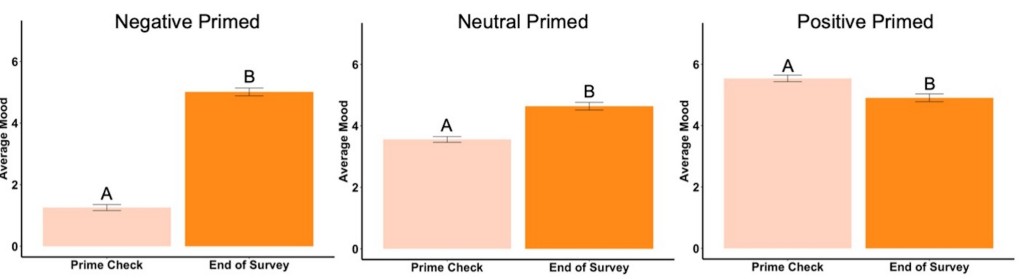

**Figure 3  Prime check mood and end of survey mood in Experiment 1.** Note: Different letters denote significant differences between the prime check and the end of survey mood check. Error bars show standard errors.

## Demographics

Overall demographics (age, dog experience, ethnicity, and gender) did not significantly affect the beginning prime mood check ($F_{49,288} = 1.02$, $p = .45$), the average valence ($F_{49,290} = 1.20$, $p = .19$) or arousal ($F_{49,290} = 1.00$ , $p = .48$) reported for the dogs in the videos across the differing prime groups.

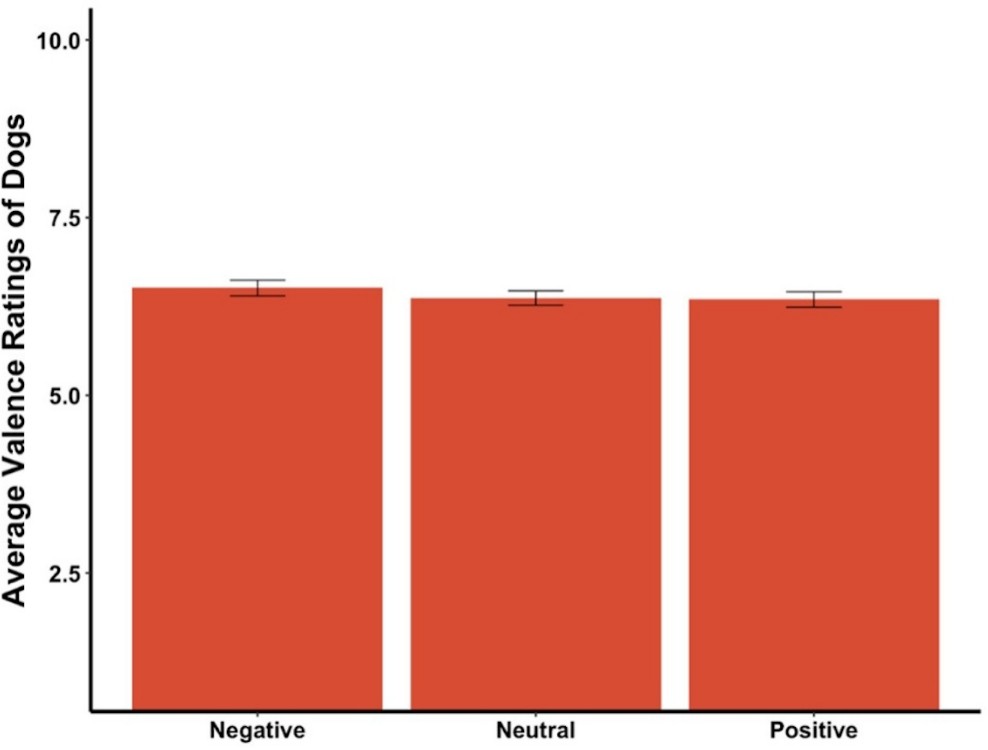

**Figure 4** **Average valence ratings in Experiment 1.** Note: Error bars show standard errors.

### Valence and arousal

The average valence reported for the dogs in the videos was not affected by the priming condition ($F_{2,290} = 0.66$, $p = .52$, Fig. 4). The average valence reported was not significantly different between the priming conditions for the first video, the first three videos, videos where the dog was in a negative state, a neutral state, or a positive state (all $ps > .05$).

The average arousal reported for the dogs in the videos was not affected by the priming condition either ($F_{2,290} = .67$, $p = .51$). Similarly, the average arousal reported was not significantly different between the priming conditions for the first video, the first three videos, videos where the dog was in a negative state, a neutral state, or a positive state (all $ps > .05$).

### Free responses

There were no statistically significant differences for all coded free responses ($p > .05$).

## DISCUSSION

These results challenged our expectations and the well-documented phenomenon of emotional congruence in priming. The prime check was significant, showing we had induced different moods in the three groups; however, these moods did not affect perception of dog emotions. Valence and arousal ratings of the dogs in the videos were the same across the three groups.

Prior research has consistently demonstrated that mood influences emotional perception. For instance, one study found that German-speaking participants primed into one of three different emotional states by watching video clips—negative, neutral, or positive—exhibited mood-congruent biases. Specifically, those in the positive group were more likely to classify faces as happy, while participants in the negative group tended to classify the same faces as sad (*Trilla, Weigand & Dziobek, 2021*). A similar effect was found in a participant sample of Chinese young adults (*Lee et al., 2008*). *Gerger, Pelowski & Ishizu (2019)* found that abstract stimuli patterns were rated as more positive following exposure to positive scenic imagery primes and more negative following negative scenic imagery primes. Similarly, *Steephen et al. (2021)* reported that positive moods induced by watching movie scenes led to happier interpretations of happy, neutral, and ambiguous expressions, while negative moods led to less positive ratings. Therefore, while this emotional congruence effect (*Niedenthal & Setterlund, 1994*) was expected, we found no effect of mood on the valence and arousal ratings of dog emotions, whether they were in negative, neutral or positive states.

One possible explanation may lie in the priming materials themselves. Previous work has highlighted the importance of the visibility and clarity of priming stimuli. *Banse (2001)* found that when priming images were explicitly visible, participants exhibited the expected emotional congruence effect. However, when the images were subliminally presented (masked), an opposite effect emerged, where participants responded in a way that contrasted with the emotional tone of the prime. This suggests that the nature and salience of the priming materials can critically influence the direction of emotional perception. It may be plausible that our priming materials did not align well with participants' expectations or emotional processing mechanisms, thus leading to inconsistent effects. However, the NAPS photo database has been widely used, cited and validated across multiple studies (*Marchewka et al., 2014*; *Riegel et al., 2016*), and therefore, this explanation for our own findings is unlikely.

We also found an increase in mood across the experiment for those in the negative and neutral mood groups, and a decrease in mood for those in the positive mood group. It is somewhat surprising that watching videos of dogs on a black background, of which only one third were in putatively positive contexts, lifted the mood of participants in the negative and neutral groups. However, without an initial mood check question, we cannot be sure whether these participants were simply returning to their baseline mood at the end of the survey.

## EXPERIMENT 2
## MATERIALS & METHODS

To further investigate the surprising results of Experiment 1 and this potential species-specific dimension of priming, we designed a second experiment. This time we deliberately used dog images as priming materials in order to examine whether species-specific priming materials could elicit a detectable emotional response in the participants. By building on the ambiguous findings of the first experiment, the second study aimed to explore whether

cross-species priming mechanisms differ fundamentally from human-centric models of emotional congruence. This adjustment not only sought to clarify the role of priming materials but also introduced an innovative avenue for understanding how humans perceive and interpret emotions in animals. We also implemented an initial mood check question in the beginning of the survey in order to get an indication of the participants' initial mood and compare that to the end of the survey. This second mood measurement was included specifically to examine the temporal stability of the induced emotional state, as mentioned in the discussion for Experiment 1 above.

### Methods

The methods of Experiment 2 were the same as Experiment 1 with the following exceptions.

### Priming photos

Images of dogs selected from the Open Affective Standardized Image Set (OASIS) were used to induce general positive, neutral, or negative emotional states in human participants taking this survey (*Kurdi, Lozano & Banaji, 2017*).

### Survey

The same survey was used as in Experiment 1 with the following exceptions: The first question asked how participants were feeling on a scale from 1–7. This was a change from Experiment 1, to obtain an assessment of participant mood prior to mood induction, as well as after. After demographic questions, the priming photos were images of dogs.

### Participants

A new sample of Arizona State University Psychology undergraduates participated in this study. ASU's Institutional Review Board for human subjects approved this extension on 2/22/2024 (STUDY00018423). The survey was launched on 3/05/2024 and ended when 300 respondents had completed the survey, 100 per each emotional condition version of the survey.

### Free response coding

The same two undergraduate assistants coded this experiment's free responses, still unaware of the study's aims. Cronbach's alpha again showed high consistency of interrater reliability across all subcodes (valence, $\alpha = .980$; arousal, $\alpha = .983$; anthropomorphism, $\alpha = .839$; mentalizing, $\alpha = .940$; action, $\alpha = .937$). The two coders then individually scored all 2,664 free responses for the five dimensions listed above.

## RESULTS

### Response inclusion criteria and demographics

A total of 100 participants took each of the surveys. Once those who did not answer the check in question were removed from the dataset, there was a total of 100 for the negative priming group, 99 for the neutral and 97 for the positive.

Demographic descriptions are in Table 1 for the three prime group condition surveys. For analysis, age class, gender, ethnicity, and dog experience were combined into the same groups as Experiment #1.

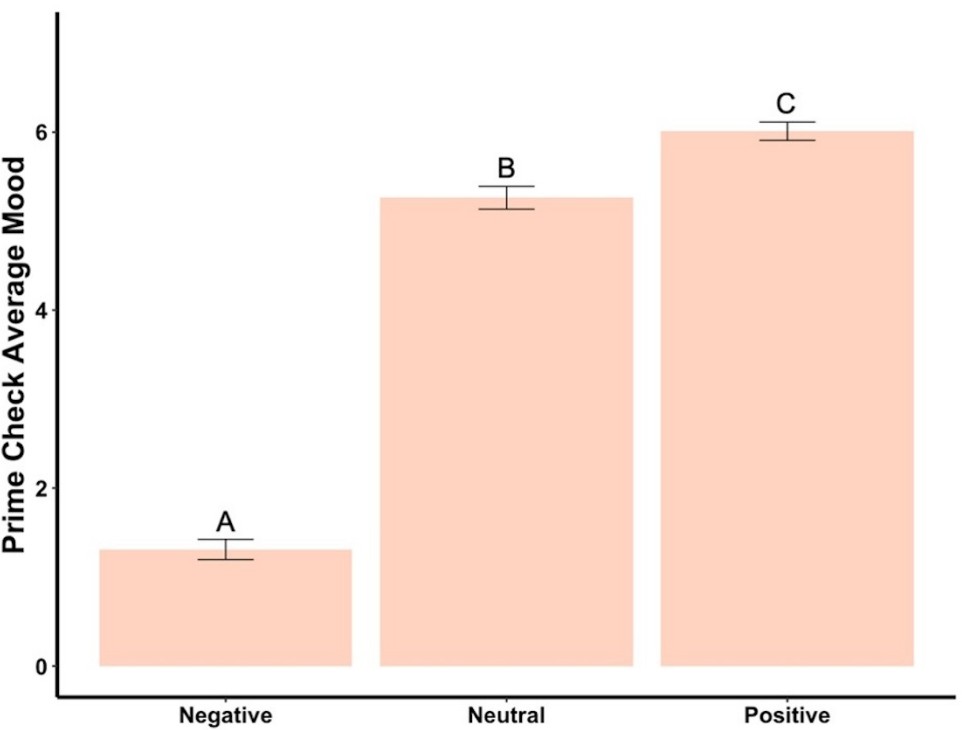

**Figure 5  Prime check mood averages across the three primed groups—Experiment 2.** Note: Different letters denote significant differences between conditions. Error bars show standard errors.

## Beginning, priming and ending moods

The beginning mood was not statistically different among the three groups ($F_{2,295} = 1.80$, $p = .17$). However, after seeing the mood priming photographs, the mood induction was successful, with each group reporting statistically significantly different moods ($F_{2,295} = 480.54$, $p < .001$, $\eta p^2 = .77$, Fig. 5). *Post hoc* tests revealed each group was significantly different from the others, with negative having the lowest mood and positive the highest. Ending mood was not statistically different among the three groups ($F_{2,295} = 0.54$ , $p = .58$).

In the negatively primed group, the beginning mood was significantly higher than the prime check mood ($t_{99} = 21.99$, $p < .001$, $d = 2.20$, Fig. 6) but there was no difference between the beginning mood and the ending mood ($t_{99} = .09$, $p = .93$). The prime check mood was significantly lower than the end mood ($t_{99} = -23.15$, $p < .001$, $d = 2.32$, Fig. 6).

In the neutrally primed group, the beginning mood was significantly lower than the prime check mood ($t_{98} = -4.17$, $p < .001$, $d = 0.42$, Fig. 6) and significantly lower the ending mood ($t_{98} = -2.46$, $p = .02$, $d = 0.25$, Fig. 6), meaning the participants' mood increased from the beginning to the end of the survey. The prime check mood was also significantly higher than the end mood as well ($t_{98} = 2.40$, $p = .02$, $d = 0.24$, Fig. 6).

In the positively primed group, the beginning mood was significantly lower than the prime check mood ($t_{96} = -11.86$, $p < .001$, $d = 1.20$, Fig. 6) and significantly lower the ending mood ($t_{96} = -3.76$, $p < .001$, $d = 0.38$, Fig. 6), also meaning the participants' mood

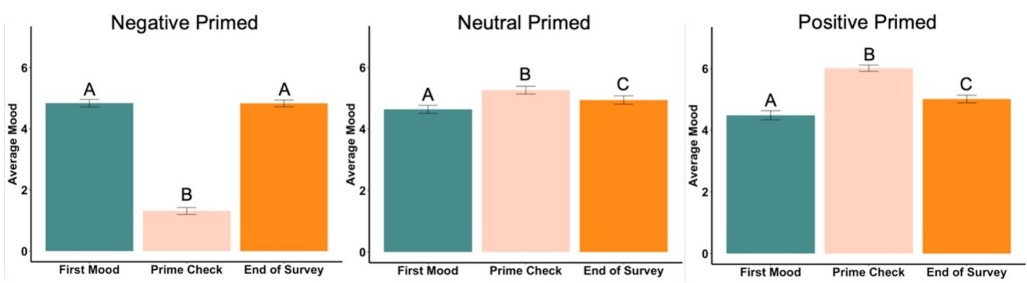

**Figure 6** **First mood, prime check mood and end of survey mood in Experiment 2.** Note: Different letters denote significant differences. Error bars show standard errors.

increased from the beginning to the end of the survey. In addition, the prime check mood was significantly higher than the end mood ($t_{96} = 7.23$, $p < .001$, $d = 0.73$, Fig. 6).

## Demographics

Overall demographics (age, dog experience, ethnicity, and gender) did not significantly affect the beginning prime mood check ($F_{41,295} = 0.91$, $p = .62$), but did affect the beginning mood ($F_{41,295} = 1.47$, $p = .04$), although further *post hoc* tests did not reveal any other significant individual predictors. Demographics (age, dog experience, ethnicity, and gender) did not significantly impact the average valence ($F_{41,295} = 0.94$, $p = .59$) or arousal ($F_{41,295} = 1.29$, $p = .12$) reported for the dogs in the videos across the differing prime groups.

## Valence and arousal

The average valence reported for the dogs in the videos was affected by the priming condition ($F_{2,295} = 4.35$, $p = .01$, $\eta p^2 = .03$, Fig. 7). *Post hoc* tests revealed significant differences between participants in the positive primed group and those in the negative group, in that participants in the positive group rated the dogs as more negative while those in the negative group rated the dogs as more positive. There was a statistically significant difference among the average valence reported for videos of the dogs in neutral conditions ($F_{2,295} = 4.15$, $p = .02$, $\eta p^2 = .03$,), with participants in the positive group rating the neutral videos of dogs more negatively while the negative group rated the neutral videos of dogs more positively. There was also a significant difference among average valence of the first three videos, which were all of the dog Canyon ($F_{2,295} = 3.78$, $p = .02$, $\eta p^2 = .03$), where participants in the positive group rated the videos as more negative while those in the neutral group rated the same videos as more positive. There was a marginally significant difference between average valence of the positive videos among the primed groups ($F_{2,295} = 2.83$, $p = .06$, $\eta p^2 = .02$), where those in the positive group rated the videos as more negative while those in the negative group rated the videos as more positive. The average valence reported was not significantly different between the priming conditions for the negative dog videos, the first video only, the middle three videos of the dog Henry, or the last three videos of the dog Oliver (all $ps > .05$).

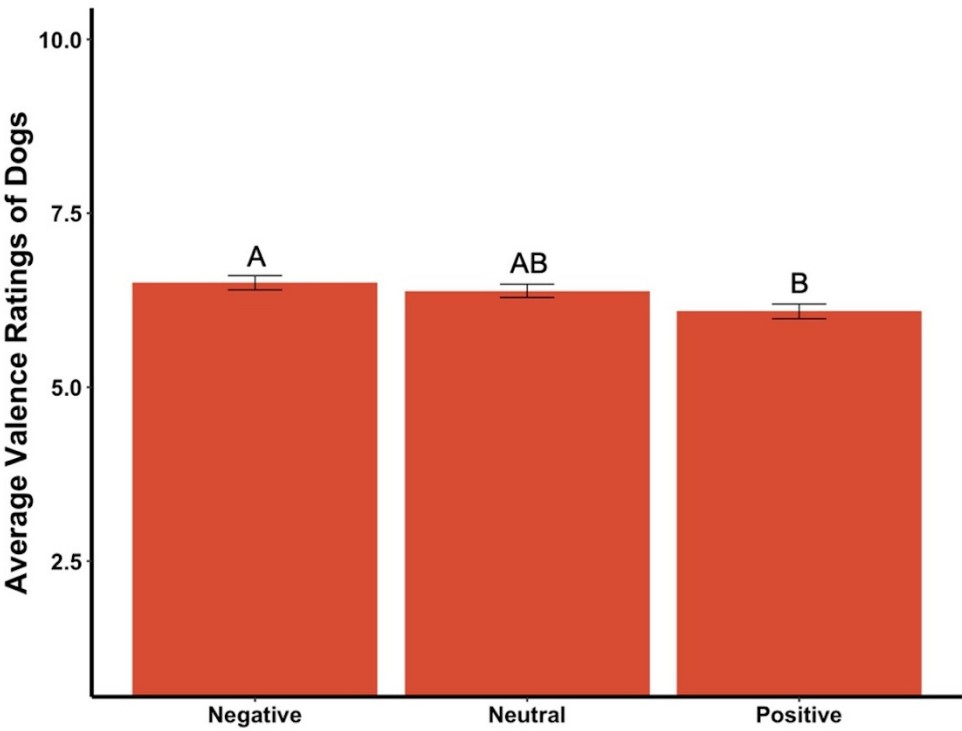

**Figure 7  Average valence ratings in Experiment 2.** Note: Different letters denote significant differences. Error bars show standard errors.

The average arousal reported for the dogs in the videos was not affected by the priming condition ($F_{2,295} = 0.16$, $p = .85$). Similarly, the average arousal reported was not significantly different between the priming conditions for the first video, the first three videos, videos where the dog was in a negative state, a neutral state, or a positive state (all $ps > .05$).

## DISCUSSION

In Experiment 2, we saw an effect of mood on the perception of dog emotions, but in the opposite direction than predicted. As in Experiment 1, the prime was effective, in that the three groups were induced into distinct moods. However, this time those in the negative mood group rated the dogs as more positive, while those in the positive mood group rated the dogs as more negative. This was a surprising finding, as it indicated an emotional contrast effect, rather than emotional congruence.

While mood effects on emotional perception have never been studied before with animal stimuli, some studies of the perception of human emotions do indicate a contrast effect. One study found that when participants were made aware of the priming mood induction, showcasing how one's appraisal of another's emotional state stems from both a feeling and knowledge component, emotional contrast occurred (*Neumann, Seibt & Strack, 2001*). In this study, it was first found that participants exposed to a cheerful voice reported finding a

cartoon more amusing compared to those placed in a bad mood by listening to a sad voice. However, when attention was drawn to the mood induction procedure, this trend reversed, with sad participants rating the cartoon as funnier. Similarly, *Banse (2001)* cited familiarity with priming materials as important for people implementing either the contrast effect or congruence in emotional perception. This seem unlikely as an explanation for our findings because we did not draw participant's attention to the mood induction procedure.

*Manierka et al. (2021)* found that naturally occurring positive moods were associated with increased recognition of scared faces, but worse recognition of happy faces, suggesting that moods artificially induced in the laboratory may have different impacts on emotion perception than natural moods. This also does not seem a plausible explanation for our findings because our study did not rely on naturally occurring moods.

In a study of the perception of emotions in virtual people, neutral characters presented in a negative context were perceived more positively than those in a positive context (*Cartaud, Lenglin & Coello, 2022*). In the present study we did not manipulate context so this, too, is not viable as an explanation for our findings.

*Hsu & Wu (2020)* emphasized that there can be sequential effects in the categorization of emotional expressions. In their study, participants were required to categorize facial expressions that were part of a sequence. Researchers manipulated the emotional similarity of successive stimuli and explored whether participants' judgments aligned with the prior expression (congruence effect) or diverged from it (contrast effect). They found that participants might categorize similar-looking stimuli together, resulting in an emotional congruence effect. Conversely, when two stimuli appear markedly different, participants were more inclined to perceive them as belonging to distinct categories, leading to a contrast effect. However, in our experiments, participants were primed with emotionally charged dog images before viewing subsequent dog-related content, yet contrast effects still emerged even though the stimuli shared clear emotional and categorical similarities. This contradicts the predicted congruence effect based on sequential similarity.

## General discussion

In two experiments we have illustrated how a person's current mood may influence their perception of emotions in dogs. Notably, this effect was specific to the species of the mood-inducing materials. When individuals were exposed to images of humans and human-related objects, their mood was altered as predicted, but it did not significantly impact how they perceived dog emotions. However, when their mood was primed by presentation of images of dogs, this significantly altered their emotional interpretations, but in the opposite direction from that typically observed. In our study, negative images led to more positive reported affect and vice versa. Additionally, we discovered that watching videos of dogs—even if the dogs are shown in neutral or negative affective states—boosted human mood. Overall, this study introduces a novel approach to understanding emotion perception in dogs and, potentially, animals more broadly.

The failure of the moods induced in Experiment 1 to yield any significant impacts on the valence and arousal ratings of the dogs in the videos is surprising in light of previous research

on mood induction, which typically demonstrates that mood significantly influences the perception of emotions in others.

Some prior studies implicate the intensity of mood induction in the production of impacts of mood on emotional perception. For example, *Van Der Does (2002)* investigated the impact of negative affect on cognitive reactivity for depressed individuals using both music and film. He found that the music prime was more effective and, in turn, had a larger impact on the cognitive reactivity score. However, another study (*Storbeck & Clore, 2008*) induced participants into either happy or sad moods in order to test the impact of mood on memory recall. These authors noted that, although participants differed in the intensity of mood induction, there was no cutoff point at which mood priming became ineffective. *Diener, Cha & Oishi (2023)*, highlighted that most mood induction studies operate within mild mood intensities, with extreme mood states being largely absent.

We compared overall mood between Experiments 1 and 2. For those in the negative prime groups, there was no difference between the mood reported after seeing the prime photos between Experiments 1 and 2 ($t_{192} = -.363$, $p = .72$). However, for those in the neutral and positive prime groups, mood was statistically significantly higher in Experiment 2 than in Experiment 1 ($t_{194} = -10.71$, $p < .001$ for the neutral group; $t_{192} = -2.99$, $p = .003$ for the positive group). End mood did not significantly differ between all six different versions of the survey taken across both experiments ($F_{5,586} = 1.25$, $p = .29$) and there were no statistically significant differences between average valence or arousal for any groups between experiments one and two (all $ps > .05$).

Given these findings of only limited differences in the magnitude of mood induction between our two experiments and the findings of *Storbeck & Clore (2008)* and *Diener, Cha & Oishi (2023)* that magnitude of mood induction is not crucial for an impact of induced mood on subsequent emotion judgements, the difference in mood intensities between Experiments 1 and 2 does not provide a compelling explanation for the observed difference in outcomes between those experiments.

Although contrast effects have been observed in prior studies on human emotion perception, none appear to align fully with the findings from Experiment 2. Previous research has shown that emotional contrast effects can occur when negative primes make subsequent neutral stimuli appear more positive and vice versa. This phenomenon is typically explained by emotional shifts or category-based comparisons triggered by the priming process. However, the contrast effect found in our experiments appears specific to dog-related priming stimuli, with no parallel effects observed when using human-related primes. Previous research has also explored when people exhibit the contrast or congruence effect in relation to using past information, rather than examining how these effects are influenced by prior mood. *Schwarz & Bless (1991)* reviewed studies showing that congruence effects occur when information is included within a category, causing a target to appear more similar to the category, while contrast effects arise when information is excluded, making a target seem more distinct from that category. However, this framework does not fully explain our study's findings. When participants were shown images of dogs and asked to evaluate canine emotions—a case where the target closely aligns with the category—we observed contrast effects instead.

We are aware of only one prior study that has investigated priming of mood and emotion with dog-related materials. *Wells, Morrison & Hepper (2012)* investigated priming of stereotypes relating to the German shepherd dog breed. In this experiment there were four different groups: those exposed to a positive story of a German shepherd saving a person, those exposed to a negative story of a German shepherd harming a child, those exposed to positive images of a German shepherd looking 'friendly,' and finally those exposed to negative images of a German shepherd looking 'aggressive.' After exposure to the priming materials, participants were asked to look at photographs of dogs belonging to five different breeds (including the German shepherd) and rank them on five traits (approachable, friendly, intelligent, aggressive and dangerous). Wells et al. found that participants reported impressions of those dogs congruent with the priming materials to which they had been exposed. Those who were exposed to both the negative story and negative images of the German shepherd rated the dog breed as less approachable, more aggressive and more dangerous than those exposed to the positive primes. This study indicates that emotional priming with dogs specifically does have an effect on our perceptions of dogs but reports emotional congruency, rather than the emotional contrast that we observed. This study differs from ours in multiple ways, however. In particular, we did not ask about specific dog qualities, but rather how dogs appeared to be feeling. Therefore, while showcasing that using dog images or media to emotionally prime people can be effective, it still does not offer an explanation for our current findings.

Therefore, to our knowledge, our study is the first to look specifically at how mood induction with dog specific priming imagery affects emotional perception of dogs and showcases how this differs in comparison to mood induction with human related priming imagery. The departure from established patterns of congruent priming towards contrasting priming suggests that our study may have uncovered a novel dynamic, potentially linked to the unique characteristics of the stimuli used and the context of cross-species interactions.

We also found that participants' mood increased after watching the videos of dogs they were asked to rate. While there is plentiful of research that showcases that interacting physically with dogs increases mood (*e.g.*, *Peel, Nguyen & Tannous, 2023*; *Picard, 2015*), prior studies have also shown that even watching videos of dogs can increase mood. One such study had participants watch videos of a dog playing and being active or lying down on a bed and found that mood increased more in response to the dog videos than when watching videos of waterfalls and running streams (*Ein, Reed & Vickers, 2022*). Another study found increased mood from watching social media pet videos (*Zhou, Yin & Gao, 2020*). However, we find it intriguing that our study indicates that merely watching videos of dogs against a black background—even when some are shown in putatively negative moods—can still elevate participants' moods. This elucidates the powerful positive effect that dogs have on our own emotional wellbeing.

There were no significant results from the free response analysis. The lack of significant results from the free responses, particularly in Experiment 2 where there was significant impact of the experimental manipulations on the scaled valence responses, could indicate that people freely describe dog emotions differently, or less sensitively, than when they rate them on a numerical scale. Using mixed methods (both quantitative and qualitive data)

can have important impacts as shown here, in that the way in which people report and perceive dog emotions is also dependent on how they report it. This shows it should be used in future studies (*Almalki, 2016*).

This study was limited by the use of an undergraduate student population. However, this subject pool enabled us to utilize a large sample which, while young and about 50% White, still represented a more diverse population than many previous studies of human perception of dog emotions that have relied on snowball sampling among the authors' acquaintances, leading to samples skewed for their enthusiasm for and experience with dogs. Another important limitation of the current study is the restricted set of dog stimuli used—only nine videos featuring three individual dogs, each expressing one of three emotional states. While care was taken to select representative and naturalistic examples, the limited diversity of dogs and expressions means that our findings may not generalize to a broader range of canine appearances, behaviors, or emotional displays. Future studies should replicate and extend this work using a larger and more varied set of stimuli to determine the robustness and boundary conditions of the observed effects.

## CONCLUSIONS

Exploring the questions raised by the findings here suggests myriad future studies that may offer promising insights into the complexities of emotion perception across species and contexts. For example, future research could compare mood priming effects with cats and dogs on the one hand and wild animals on the other, to see whether the effect is restricted to domesticated animals. Furthermore, future research could explore differences in priming with imagery of wolves and coyotes, to see whether there is a specific effect due to canids. Studies could also investigate priming with non-animate actors, such as robots or cartoon characters. These lines of inquiry would expand on existing knowledge by examining not only interspecies differences but also the evolutionary and adaptive mechanisms shaping human-animal emotional interactions. Additionally, investigating cultural factors that influence how people interpret animal emotions could reveal whether societal norms and experiences affect the congruence or contrast observed in mood priming effects.

This research sheds light on how humans perceive and interact with animals through emotional priming and extends our understanding of multispecies relationships. By exploring transspecies emotional priming, we gain insights into how animal imagery influences human emotions and perceptions, with implications in therapeutic and educational settings. This line of study not only deepens our knowledge of emotional perception mechanisms beyond the human domain but also emphasizes the significance of species-specific emotional dynamics. These discoveries emphasize the importance of considering species-specific priming effects in emotion research and hold real-world implications for how we perceive, empathize with, and care for dogs and other animals.

## ACKNOWLEDGEMENTS

Big thanks to the dogs and their pet parents who were amazing actors and subjects for our videos: Danny Jackson and Canyon, Rachel Brown and Henry, and Rich Molinaro and

Oliver. Thanks also go to Dr. Holly Miller at General Mills for her guidance and ideas for this project. Finally, we want to deeply thank Gabrielle Bucci and Sara Bojczuk for coding all free responses.

### Funding
The authors received no funding for this work.

### Competing Interests
The authors declare there are no competing interests.

### Author Contributions
- Holly G. Molinaro conceived and designed the experiments, performed the experiments, analyzed the data, prepared figures and/or tables, authored or reviewed drafts of the article, and approved the final draft.
- Clive D.L. Wynne conceived and designed the experiments, authored or reviewed drafts of the article, editing, and approved the final draft.

### Human Ethics
The following information was supplied relating to ethical approvals (i.e., approving body and any reference numbers):

Arizona State University Psychology undergraduates served as participants for this study. ASU's Institutional Review Board for human subjects approved this research project on 8/18/2023 (STUDY00018423).

### Animal Ethics
The following information was supplied relating to ethical approvals (i.e., approving body and any reference numbers):

This research was approved by Arizona State University.

### Ethics
The following information was supplied relating to ethical approvals (i.e., approving body and any reference numbers):

Arizona State University Psychology undergraduates served as participants for this study. ASU's Institutional Review Board for human subjects approved this research project on 8/18/2023 (STUDY00018423).

### Data Availability
The raw data are available in the Supplemental File.

### Supplemental Information
Supplemental information for this article can be found online at http://dx.doi.org/10.7717/peerj.20411#supplemental-information.

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
