# Peer review of "Paw-spective shift: how our mood alters the way we read dog emotions"

_PeerJ, doi:10.7717/peerj.20411_

## Round 0.1 · original submission · Major Revisions

· Academic Editor

Major Revisions

First, my apologies for the delay in getting an editorial decision rendered here, and thank you for your patience. I’ve managed to get 3 solid reviews on your paper, most of which are quite positive. I’ve also read the paper with considerable interest, and I’m happy to invite a revision.

So where are we at?

Rev1 asks for some clarification and justification that explicit measures of mood are appropriate here.

Rev2 has some statistical concerns, requests a fair bit of methodological clarification, and offers a few other critiques.

Rev3 also has some points of clarification requested, some questions about the data, and some concerns to address regarding how the data are being interpreted (in light of your experimental design).

All told, I’m happy to invite a revision. I think that most – if not all – of these concerns can be addressed in a revision, but it’s likely that I’ll send the paper back to at least one or two of the reviewers to have them take a second peek.

Reviewer 1 ·

Basic reporting

The only place in which I think some further explanation, or at least a citation, would be helpful is to back up the claim that explicit measures of mood, such as the Likert scale used in this study, do capture the kind of processes that influence other judgements in the way this study requires. Explicit measures of moods/emotions can be a bit tricky for a few reasons, but in this case, in particular, it seems that the influence a mood has on the identification of a mood in a dog in a video would be implicit. As such, it might be different from the explicit report. I gather that these "prime checks" do work, based on other work on moods that the manuscript cites. However, a citation where this is introduced would help (Methods & Materials for experiment 1; lines 175-176).

Experimental design

-

Validity of the findings

-

Reviewer 2 ·

Basic reporting

This is an interesting study on how humans’ affective states might affect their interpretation of dog emotions. The findings are somewhat surprising and therefore, intriguing. The paper is generally well written. The methodology is generally strong, although I do have some concerns. In fact, I think the entire statistical approach should be revisited, which may result in a different pattern of findings.

I have the following minor comments:
Please specify the source of comparisons so that there is no room for ambiguity. For example, on line 74, state that you mean “than participants not in a sad mood” rather than “than facial expressions of other moods.”
It would not hurt to be more explicit about why you think that perceiving animal emotions assists with their welfare, even if you think it may be obvious.
There is a reasonable literature on how owner attachment predicts greater attribution of especially more complex emotions to companion animals that could be cited here.
Lines 80-81, if I am not mistaken, Walker et al. showed that women were more likely to perceive that their companion animals were capable of expressing grief or love, but the way this sentence is worded, it is misleading. It reads as if women were able to accurately perceive love or grief, but I don’t think that was tested.
Please be clear if each dog was recorded expressing each of the three emotions (rather than “or”) (lines 118, 123).
You do not need to create three separate surveys. You could randomly assign 1/3 of the participants to each of the blocks where the mood manipulation is contained.
What were the end points of the Likert scale to assess mood? Did you only assess mood as negative or positive? You might consider using the PANAS, where participants rate how much they feel each of various emotions.
Why were the dog videos not presented in random order across participants?
How did you settle on 300 participants? There should be an a priori power assessment.
Did you include any attention checks or otherwise include data cleaning criteria?
Line 146, 163, there “were”
Line 23 missing ‘ after participants
Line 47, change “while” to “whereas” or “although” and check for other non-temporal uses of while and since (e.g., line 77).

Experimental design

Your statistical approach seems a little piecemeal. Why not use a mixed model where you can assess the change in mood as a function of condition as well as participant attributes? Alternatively, it is a bit debatable whether you should analyze the effect on attributed mood state to the mood induction manipulation or the participants’ actual mood when evaluating the dog videos. You could run a mediation model looking at whether any effects of the condition on rating were mediated through the participants’ actual mood.
Why did you use a free response of how the dog was feeling rather than having participants choose from among alternatives?
Why did you split participants by age rather than treating age as a continuous variable/covariate?
I think you mean you dummy-coded rather than “combined” people into groups based on other demographic variables.
Ideally, you would have measured mood before and immediately after the mood induction procedure. You need to show that your mood induction changed the mood to verify that it worked. It is possible, although unlikely, that participants in your positive affect group were already experiencing a more positive mood, for example. In any case, showing that moods differ after is suggestive that the induction worked, but it can’t entirely validate it (line 250). Did you add this pre-mood measure in Study 2? If so, you should emphasize that this is a change from Study 1 to improve on the methodology.
I don’t follow the logic that participants would need to be primed with stimuli congruent with the objects to be rated to show the emotion-congruency effect. If the point is that when I’m in a positive mood, I’m more likely to attribute positive emotions to others than when I’m in a negative mood, it shouldn’t matter how I came to be in a positive or negative mood.
You should report confidence intervals and effect sizes.
From the way you have written your results, it isn’t clear which result is tied to which analysis. The entire format for reporting results is not standard.
In Study 2, did you have an interaction of dog or dog emotion condition and manipulation group? Because otherwise, why do you separately analyze the rating of the neutral videos alone and the first three videos? I don’t follow where these analyses come from if not based on interactions, which were not reported.
You should include in a limitations section the fact that you used only 9 images of dogs, with only 3 dogs expressing each of 3 emotions. Therefore, it is quite possible that your findings would not generalize to a broader pool of stimuli. Caution should be used when considering that you have uncovered a novel effect (e.g., line 565).
It doesn’t seem like you controlled for gender, age, and experience with dogs in your main analyses, but you should.

Validity of the findings

I think the findings may change when the analyses are revised.

·

Basic reporting

Very well written.
One grammatical error noted on line 183 ("was done ensure" should be "was done to ensure")
Thorough literature review. Most every time I had a question an answer was provided from previous research listed and described by the authors. Very comprehensive and well done.

Suggest that the authors provide a short but more detailed description of the terms "valence" and "arousal." I was uncertain if body movement/activity of the dogs was considered within the category of "arousal."

**I did not see "Table 1" included in the submitted materials?? (Demographics of human participants.)

Experimental design

Very interesting well-defined and described research.
I commend the authors for taking on this research which is valuable but difficult to design and conduct because of the many variables; not the least of which is that one of the species involved is non-verbal - they cannot be asked how they are feeling.
I deeply appreciated the description of the dogs, and the pictures included. I believe this is very important aspect of the research and research design. It is frustrating that many research manuscripts relating to the human-canine bond provide little to no information on the canines that participate in the research. It was refreshing to see that this manuscript provided great detail on the canine participants.

With respect to the canine participants, I have several questions/thoughts for the authors to consider/answer. They are listed below.
1) It may be prudent in the future to use canine participants that are all the same size, fur length/type, color, and especially breed (and maybe even gender.) Fur color, dog size, and fur length may influence the human participant's mood. Perhaps a very well-known breed such as a yellow Labrador Retriever. Also, even if the breed is the same, choose dogs to participate in the research that are very expressive regarding their mood - some dogs are more subtle with respect to their facial expressions/body language and mood.
2) No disrespect to Canyon, however, his/her dark fur and mostly dark face against the solid black background may have made it especially difficult for the human subjects to accurately see and interpret his emotional state from his/her facial expression. A highly contrasting background in relation to the dog's coat color might be a better choice.
3) Since you kept any sound that came from the dog in the video, you should address if any of the dogs were particularly vocal and if this could have influenced the results. It may be better in the future to eliminate all sound from the video so that the human participants are determining canine mood solely on visual cues from the dog.
4) You analyzed "arousal" of the dogs but it is unclear if this includes body movement. It would be helpful to clarify this. A dog that is more active may be perceived by human participants as being in a particular mood.

With respect to the human participants, I have several questions/thoughts for the authors to consider/answer. They are listed below.
1) It may be important for the participants to self-identify if they have a condition such as Autism Spectrum Disorder or ADHD which may inhibit their ability to discern emotion from facial/body language. Not sure if this is being too picky, but I offer it for your consideration.
2) I did not see Table 1 (human demographic information) included with the submitted materials?? This is important information to include in the paper.
3) Ages of the human participants were grouped into 18-21 years and > 22 years. I was curious how many participants were in a different generational category. It has been reported that Gen Z tends to lack face-to-face communication and has an attention span ~8 seconds. I wonder if their exposure and dependence on social media has impaired their ability to accurately interpret moods based on facial expression and body language?? It would be very interesting to determine if different generations (in general) have different abilities to accurately detect the mood of others - including our canine companions. Is there published research on this?
4) Since I did not see Table 1, I do not know what the breakdown of gender and ethnicity was. But more importantly, I was curious about how the human participants self-categorized their dog experience level. Were each of the experience categories with dogs well-defined for the participants? I feel like experience could have been overstated by the participants. I also thought that it would be useful to determine the "interaction history" of the participants with dogs - i.e. any negative experiences (afraid/bitten), neutral, or positive experiences (love dogs to the point that they would cross a 4-lane busy street to go see a dog.) Past interaction with dogs is different than having experience with dogs (expertise with canine care/behavior, etc.)

Validity of the findings

While the findings did not support the hypothesis, the authors did well in exploring and describing possible reasons for the unexpected results.
It was very interesting to see in Experiment 1 that the mood of human participants in a negative and neutral state increased irrespective of the fact that 2/3 of the videos viewed were of dogs in a neutral or negative state. The authors noted the powerful positive effect of dogs on human emotional well-being (I tell my students this is the "dog-effect") and this may support E.O. Wilson's biophilia hypothesis...that the affinity humans have for dogs has an evolutionary, genetic basis. While it was unfortunate that there was no initial mood check completed by the human participants in Experiment 1, the authors addressed this in Experiment 2.
In Experiment 2 another very surprising result was that human participants with induced negative moods rated the dogs as more positive while the human participants in a positive mood rated the dogs as more negative. While some of the pictures used to induce particular moods in the human participants were rather subtle images of negative (i.e. the broken leg), neutral (military plane), and positive the data clearly showed that the difference moods were induced in the human participants. This data could be interpreted that while moods in the human participants can be effectively induced, the ABILITY of the participants to accurately perceive the moods of canines is not well developed.

Additional comments

I am curious to know if mood induction in human participants using live interactions between the participant and the induction mechanism (i.e. a live human pretending to be extremely upset/crying) can produce more intense and/or longer lasting moods??

I would suggest the authors provide a short description regarding the different dog's "ability" to clearly demonstrate overt emotions of negative and positive. i.e. did some of the canine participants have more subtle expression of emotions.

Very interesting research. It was a pleasure to read and review.

---

## Round 0.2 · Minor Revisions

· Academic Editor

Minor Revisions

Thanks for your careful revisions of this manuscript. Rev1 is clearly pleased, and Rev2 is mostly pleased as well. There are a few things for you to tidy up and/or respond to. I won't bother sending the paper back out for review again provided these things are addressed or a cogent argument is made that they are not necessary.

Reviewer 1 ·

Basic reporting

no comment

Experimental design

no coment

Validity of the findings

no comment

Additional comments

I have no further concerns.

Reviewer 2 ·

Basic reporting

I appreciate the opportunity to review this revision. The paper reads well and the study is generally conducted in a rigorous way. I think the results are intriguing and novel and deserve to be published. I just have a few minor comments.

Experimental design

I disagree with the authors’ concern about treating age as a continuous variable. If you are not comparing different age groups, you don’t have to worry about smaller numbers of participants for certain ages. I would think you are interested in whether there is a linear effect of age, not whether there are specific differences between exact age groups. However, if you have primarily a student sample, age differences may not be that interesting anyway. I think gender is probably a more relevant variable for this study.

Validity of the findings

It is unclear why you measured mood from the mood prime check to the end of the survey. This seems to show that any effect on mood from the priming was transient. Is this what you intended to show? It makes some sense to me after reflecting that those who were induced into a negative mood started to increase mood with further temporal distance from the priming and vice versa for those induced into a positive mood. You mention the idea of participants returning to baseline by the end of the survey on line 341 and this seems like a reasonable explanation.
The fact that mood changes are similar in the neutral and positive group leads me to question the positive manipulation and its effectiveness. This reinforces my thinking that it may be more important to use the participants’ actual post manipulation mood rather than the group designation as a predictor of their ratings of dog emotions. It might be worth running some analyses just to see.
You conducted a power analysis for a moderate size effect. Do mood inductions typically result in moderate effects? Perhaps the study was underpowered to detect small effect sizes?

Additional comments

“All p” should be “all ps” throughout.
Line 565 should be “We are aware of only one prior study…”
Please replace “Since” with “Because” throughout.

---

## Round 0.3 · accepted · Accept

· Academic Editor

Accept

Thank you for the follow-up analyses and your careful response to the reviewer critiques. Congrats on the new paper.